# Inhibition of the Growth of Breast Cancer-Associated Brain Tumors by the Osteocyte-Derived Conditioned Medium

**DOI:** 10.3390/cancers13051061

**Published:** 2021-03-03

**Authors:** Tomohiko Sano, Xun Sun, Yan Feng, Shengzhi Liu, Misato Hase, Yao Fan, Rongrong Zha, Di Wu, Uma K. Aryal, Bai-Yan Li, Akihiro Sudo, Hiroki Yokota

**Affiliations:** 1Department of Biomedical Engineering, Indiana University Purdue University Indianapolis, Indianapolis, IN 46202, USA; tomohiko0122@clin.medic.mie-u.ac.jp (T.S.); sunxun@iu.edu (X.S.); fengya@iu.edu (Y.F.); liu441@iupui.edu (S.L.); misahase@iu.edu (M.H.); fanyao0124@163.com (Y.F.); zhar@iu.edu (R.Z.); hmuwudi@163.com (D.W.); 2Department of Orthopedic Surgery, Mie University Graduate School of Medicine, Edobashi Tsu 2-174, Japan; a-sudou@clin.medic.mie-u.ac.jp; 3Department of Pharmacology, School of Pharmacy, Harbin Medical University, Harbin 150081, China; liby@ems.hrbmu.edu.cn; 4Graduate School of Engineering, Mie University, Edobashi Tsu 2-174, Japan; 5Department of Comparative Pathobiology, Purdue University, West Lafayette, IN 47907, USA; uaryal@purdue.edu; 6Simon Cancer Research Center, Indiana University School of Medicine, Indianapolis, IN 46202, USA; 7Indiana Center for Musculoskeletal Health, Indiana University School of Medicine, Indianapolis, IN 46202, USA

**Keywords:** breast cancer, osteocytes, conditioned medium, Lrp5, β-catenin, IL1ra, histone H4

## Abstract

**Simple Summary:**

Breast cancer is a common malignancy in women in the US. While brain metastases frequently occur from advanced breast cancer, current treatments are of limited effectiveness. In this study, we found that osteocytes and their conditioned medium (CM) presented the tumor-suppressing capability, and the overexpression of Lrp5, IL1ra, and β-catenin enhanced their anti-tumor actions. In a mouse model, the administration of osteocytes and their CM inhibited the progression of mammary tumors and tumors in the bone and brain. The minimally invasive administration allowed tumor-suppressing factors in CM to diffuse into the brain. Besides p53 and Trail, mass spectrometry-based whole-genome proteomics revealed that extracellular histone H4 was enriched in CM and acted as a tumor suppressor. We also observed that Lrp5-overexpressing mesenchymal stem cells presented the tumor-suppressing capability.

**Abstract:**

The brain is a common site of metastasis from advanced breast cancer but few effective treatments are available. We examined a therapeutic option with a conditioned medium (CM), focusing on the role of Lrp5 and β-catenin in Wnt signaling, and IL1ra in osteocytes. Osteocytes presented the innate anti-tumor effect and the overexpression of the above genes strengthened their action. In a mouse model, the injection of their CM inhibited mammary tumors and tumor-driven osteolysis. Importantly, Lrp5- and/or IL1ra-overexpressing osteocytes or the local administration of β-catenin-overexpressing CM markedly inhibited brain tumors. In the transport analysis, tumor-suppressing factors in CM were shown to diffuse through the skull. Mechanistically, the CM with overexpression of the above genes downregulated oncogenic genes such as MMP9, Runx2, TGFβ, and Snail in breast cancer cells. Also, the CM with β-catenin overexpression downregulated CXCL1 and CXCL5 and upregulated tumor suppressors such as LIMA1, DSP, p53, and TRAIL in breast cancer cells. Notably, whole-genome proteomics revealed that histone H4 was enriched in CM and acted as an atypical tumor suppressor. Lrp5-overexpressing MSCs were also shown to act as anti-tumor agents. Collectively, this study demonstrated the therapeutic role of engineered CM in brain tumors and the tumor-suppressing action of extracellular histone H4. The result sheds light on the potential CM-based therapy for breast cancer-associated brain metastases in a minimally invasive manner.

## 1. Introduction

Breast cancer is one of the most common cancers for women worldwide and one in eight women in the US will develop breast cancer in her lifetime [1,2]. The frequent sites of metastasis for advanced breast cancer includes the bone, liver, lung, and brain [3,4], among which brain metastasis has one of the worst prognoses with an extremely low survival rate [5]. While existing treatment procedures include neurosurgery, whole-brain radiation therapy, stereotactic radiosurgery, and hormonal therapy, the severe impairment in neurocognitive ability and quality of life is commonly associated with these regimens [6,7]. Most chemotherapeutic agents are not fully effective because of the limited permeability to the brain [8]. We previously reported that the loading-driven regulation of dopamine suppressed the progression of brain tumors in a mouse model [9]. To develop a further effective therapeutic strategy, this study examined the possibility of a conditioned medium (CM)-based option using a minimally-invasive procedure. 

While astrocytes as brain-residing cells and mesenchymal stem cells (MSCs) were partially evaluated as therapeutics in regenerative medicine [10,11], the prime target herein was osteocytes and their CM since they are reported to have the tumor-suppressing capability [12]. Osteocytes are the most abundant bone cells [13,14] and in response to skeletal loading, they activate Wnt signaling and orchestrate bone remodeling [15]. Lrp5 is a co-receptor of Wnt signaling, while β-catenin is a transcription co-activator [16,17,18,19]. We have previously shown that osteocytes can suppress migratory behaviors of breast cancer cells by inhibiting an epithelial-to-mesenchymal transition (EMT) [20]. Although Wnt signaling is known to act as an inducer of tumorigenesis [16,21], its role in tumor-osteocyte interactions is elusive. In this study, we overexpressed Lrp5 and β-catenin in osteocytes and aimed to enhance their innate anti-tumor capability. Lrp5 was overexpressed in this study since we previously observed that osteocytes with a higher expression level of Lrp5 were more potent to suppress tumor progression. We also observed that the antitumor action of dopamine was mediated in part by Lrp5 together with dopamine receptors [9]. β-catenin was also overexpressed since it is elevated by Lrp5-mediated Wnt signaling [16]. We also overexpressed interleukin 1 receptor antagonist (IL1ra) in osteocytes since IL1ra is reported to reduce tumor-promoting inflammation [22,23].

One of the major challenges in the treatment of brain tumors is the delivery of therapeutic agents because of the blood-brain barrier. In this study, brain tumors were induced by inoculating mammary tumor cells to the frontal lobe of the brain, using a mouse model previously described [24]. As a novel treatment option, we first examined the cell-based treatment in which osteocytes were co-injected with tumor cells at the frontal lobe. Since osteocytes are not brain-resident cells, we next evaluated the efficacy of the cranial subcutaneous injection of osteocyte-derived CM in a minimally invasive procedure without causing any damage to the skull. The permeability of osteocyte-derived CM to the brain was determined by an ex vivo skull assay using mammary tumor cells as well as freshly isolated HER2-positive breast cancer tissue that is known to metastasize frequently to the brain [25,26,27]. 

In the proposed therapeutic option, we postulated that the tumor-suppressing CM contained an elevated level of tumor suppressors. As a candidate of tumor-suppressing proteins, we evaluated the expression levels of p53, TRAIL, LIMA1, and DSP in the CM, which are known to act as anti-tumor agents or apoptosis inducers [28,29,30,31]. We also conducted whole-genome proteomics and predicted tumor-suppressing protein candidates that were enriched in CM. One candidate, identified by mass spectrometry, was histone H4, one of the core histones to form octameric nucleosomes [32]. Histone H4 is reported to increase caspase 3 activity and induce apoptosis in neutrophils [33,34]. As tumor-promoting proteins that are expected to be downregulated in osteocyte-derived CM, we focused on the expression levels of CXCL1, and CXCL5, two chemokines that are involved in the migration of breast cancer cells [35,36]. The responses of tumor cells to osteocyte-derived CM were characterized by the expression of tumorigenic genes such as Runx2, MMP9, TGFβ, and Snail [37,38,39,40,41]. Collectively, we demonstrated that the CM, derived from genetically modified osteocytes and MSCs, can suppress the progression of tumors not only in the mammary fat pads but also in the metastasized bone and brain with atypical tumor suppressors such as extracellular histone H4.

## 2. Results

### 2.1. Suppression of Viability, Migration, and invasion of EO771 Tumor Cells by Osteocyte-Derived CM

We first examined the effect of Lrp5-overexpressing osteocytes in the tumorigenic behaviors of EO771 mammary tumor cells, using MLO-A5 osteocytes (Figure 1A,B). The result showed that A5-derived CM significantly decreased MTT-based viability of tumor cells (Figure 1C), inhibited their migration and invasion (Figure 1D,E), and suppressed the growth of tumor spheroids (Figure 1F). Notably, the observed anti-tumor effect was enhanced by the overexpression of Lrp5 in osteocytes (Figure 1C–F). Taken together, the in vitro analysis revealed that osteocyte-derived CM presented the tumor-suppressing capability and the anti-tumor action was enhanced by Lrp5 overexpression.

Before examining the effect of osteocyte-derived CM in brain tumors, we evaluated its action on the progression of mammary tumors and tumor-invaded bone degradation. In a mouse model of mammary tumors and tibial osteolysis using C57BL/6 female mice, the co-injection of osteocytes to the mammary fat pad significantly reduced the size of mammary tumors and the anti-tumor effect was strengthened by the overexpression of Lrp5 (Figure 2A). Furthermore, the daily administration of β-catenin overexpressing CM markedly shrank mammary tumors (Figure 2B). The inhibitory effect was also observed in the tumor-invaded tibia. X-ray images showed that the co-injection of osteocytes with and without Lrp5 overexpression as well as the systemic administration of β-catenin-overexpressing CM suppressed tumor-driven bone loss (Figure 2C,D). 

### 2.2. Suppression of Tumor Growth in the Brain by the Co-Injection of Osteocytes

Since in vivo analyses of mammary tumors in the mammary fat pad and tibia showed the inhibitory effect of osteocyte-derived CM, we next evaluated the action of osteocytes on brain tumors on the right side of the frontal lobe. Among the three groups, the A5 group received the co-injection of osteocytes, while the A5 + Lrp5 group received the co-injection of Lrp5-overexpressing osteocytes. The activity scores, body weight, and brain damage scores were the worst in the placebo group (Figure 3A–C), whereas they were the best in the A5 + Lrp5 group (Figure 3C). In the histological analysis, the progression of brain tumors was significantly suppressed in the A5 group and the overexpression of Lrp5 in the A5 + Lrp5 group further reduced the tumor-invaded area in the brain (Figure 3D). Collectively, the co-injection of osteocytes into the brain suppressed the growth of brain tumors and the overexpression of Lrp5 in osteocytes enhanced the anti-tumor action.

### 2.3. Anti-Tumor Effects of IL1ra Overexpression on Brain Tumors

So far, we have shown that Lrp5-overexpressing osteocytes and their CM suppressed the progression of tumors in the mammary fat pad, bone, and brain. We next evaluated the role of IL1ra (Figure 4A). In the MTT, scratch-based migration, and Transwell invasion assays, IL1ra-overexpressing CM presented significantly stronger anti-tumor effects than osteocyte-derived CM without its overexpression (Figure 4B–D; Appendix A). In a mouse model of mammary tumor in which EO771 cells were inoculated into the mammary fat pad, the co-injection of IL1ra-overexpressing osteocytes markedly reduced the size and weight of tumors (Figure 4E). Taken together, the results reinforced the anti-tumor action of osteocytes and revealed that besides the overexpression of Lrp5 or β-catenin, the anti-tumor effect was also enhanced by the overexpression of IL1ra.

To further evaluate the action of IL1ra- and Lrp5-overexpressing osteocytes, we co-transfected them in osteocytes. In the IL1ra group and the (IL1ra + Lrp5) group, the activity score and body weight were superior to those in the placebo group (Appendix A), as well as the brain damage scores (Figure 4F). In the histological analysis, the size of brain tumors in the (IL1ra + Lrp5) group was the smallest (Figure 4G). The result thus indicated that the combined overexpression of IL1ra and Lrp5 further enhanced the osteocyte-driven anti-tumor effect.

### 2.4. Anti-Tumor Effect of β-Catenin Overexpressing CM on Brain Tumors

Instead of introducing osteocytes, which are not associated with brain-resident cells, using the needle-based injection through the skull, we next examined the minimally-invasive application of β-catenin overexpressing CM as a subcutaneous injection. In vitro analyses with the MTT, scratch-based migration, and Transwell invasion assays revealed that β-catenin-overexpressing CM significantly suppressed the proliferation, migration, and invasion of tumor cells (Figure 5A–C; Appendix A). In the mouse model of brain tumors, the administration of β-catenin-overexpressing CM to the head skin resulted in better activity scores, higher body weights, and improved brain damage scores than the placebo (Figure 5D–F). Histological analysis revealed that the growth of brain tumors was significantly reduced by the application of osteocyte-derived CM and the overexpression of β-catenin further reduced the tumor-invaded area in the brain (Figure 5G). No harmful side-effects, such as bleeding, inflammation, and infection, were observed at the site of CM administration. The in vivo result herein indicated that the minimally invasive injection of β-catenin-overexpressing CM was effective to inhibit the growth of brain tumors. 

### 2.5. Transport Analysis of CM Through the Skull

While the suppression of tumor growth in the brain by the administration of osteocyte-derived CM suggested the transport of tumor-suppressing factors into the brain through the skull, we examined the biophysical property of the skull and characterized the diffusive transport phenomena. The coronal histological sections showed that the murine skull consisted of a porous microstructure without any obvious penetrating hole (Figure 6A). We estimated the diffusion coefficient of bovine serum albumin (BSA) from the outside of the skull to its inside. The temporal changes in the logarithmic concentration ratio, which is defined in equation 4 in the Appendix A, were plotted as a function of time (Figure 6B). The diffusion coefficient of BSA across the skull is linked to the slope of the plots and it was estimated to be 3.30 × 10^−8^ cm^2^/s. The estimated value across the skull is approximately 20-fold smaller than that in the aqueous solution. 

To further evaluate the transport across the skull, we next conducted the MTT-based viability assay, in which EO771 cells were placed in the inside of the skull, while anti-tumor agents such as fluphenazine and β-catenin CM in its outside. The result showed that compared to the control, both fluphenazine and β-catenin CM significantly decreased the viability of tumor cells in the inside of the skull (Figure 6C). Furthermore, we conducted the same transport analysis using freshly isolated human breast cancer fragments. The result also revealed that the administration of fluphenazine and β-catenin CM in the outside of the skull significantly shrunk the tumor fragments in the inside of the skull (Figure 6D). Collectively, these in vitro and ex vivo analyses revealed that tumor-suppressing factors in CM can be transported across the skull.

### 2.6. Enrichment of Tumor Suppressors in Osteocyte-Derived CM

The tumor-suppressing action of osteocyte-derived CM to the mammary fat pad, bone, and brain indicates that tumor-suppressing proteins are enriched in osteocyte-derived CM and this enrichment is amplified by the overexpression of Lrp5, β-catenin, and IL1ra. Before examining potential tumor suppressors, we found that the levels of IL1β, Runx2, MMP9, TGFβ, and Snail in EO771 cells were downregulated by osteocyte-derived CM, and their levels were further reduced by the overexpression of the three selected genes (Figure 6E). In β-catenin-overexpressing CM, the levels of four known tumor suppressors (p53, TRAIL, LIMA1, DSP) were upregulated, whereas the levels of two tumor-promoting chemokines (CXCL1 and CXCL5) were downregulated (Figure 6F).

Consistent with the role of TGFβ and Snail that promotes tumorigenic behaviors, the administration of TGFβ elevated MTT-based viability and scratch-based motility of EO771 mammary tumor cells (Appendix A). Furthermore, the overexpression of Snail in EO771 cells promoted their viability and motility (Appendix A). 

### 2.7. Extracellular Histone H4 as a Novel Tumor Suppressor in CM

Besides the above tumor suppressors, we conducted a whole-genome proteomics analysis and identified fifty-six proteins that were enriched in β-catenin-overexpressing CM (Appendix A). We focused on the top eight candidates (Figure 7A) and examined their effect in the MTT assay. The result revealed that six out of eight candidates significantly inhibited the viability of EO771 mammary tumor cells (Figure 7B). In particular, histone H4 (H4) and ubiquitin C (Ubc) exhibited striking inhibition and we observed the elevated level of histone H4 in Lrp5-, IL1ra-, and β-catenin-overexpressing osteocyte-derived CM but not Ubc in western blotting (Figure 7C). In the MTT assay, the inhibitory action of histone H4 is stronger to tumor cells than to non-tumor cells (Appendix A). Importantly, RNA interference with histone H4 siRNA suppressed the inhibitory effect of Lrp5-overexpressing CM (Figure 7D). The treatment with histone H4 reduced the scratch-based migration and EdU-based proliferation (Figure 7E,F), while the reduction was suppressed by histone H4 siRNA (Figure 7G,H). Also, histone H4 decreased the levels of IL1α, Runx2, MMP9, TGFβ, and Snail in EO771 cells, while their decrease was reversed by the H4 siRNA-treated CM (Figure 7I). Last, the partial silencing of histone H4 downregulated the selected tumor suppressors such as p53, TRAIL, LIMA1, and DSP, and upregulated the tumor promoters such as CXCL1 and CXCL5 in osteocyte-derived CM (Figure 7J).

### 2.8. Differential Effects of Astrocyte- and MSC-Derived CM

Besides osteocytes, we evaluated the effect of astrocytes and MSCs. Astrocytes are the most abundant glial cells in the brain [42], while MSCs are commonly utilized in regenerative medicine [11]. In the MTT-based viability and scratch-based migration assays, the astrocyte-derived CM did not induce any detectable change (Figure 8A–C). In contrast to Lrp5-overexpressing osteocytes, Lrp5-overexpressing astrocyte-derived CM promoted the viability and migration of EO771 tumor cells, and the level of MMP9, Runx2, and Snail in EO771 cells were upregulated (Figure 8D). By contrast, MSCs did not present the innate anti-tumor actions but the overexpression of Lrp5 converted them into anti-tumor agents by reducing the MTT-based viability and downregulating MMP9, Runx2, and Snail, as well as N-cadherin in EO771 cells (Figure 8E,F). 

## 3. Discussion

We have presented herein that osteocytes and their CM were capable of inhibiting the progression of tumors in the mammary fat pad, tibia, and brain, and their anti-tumor capability was enhanced by the overexpression of Lrp5, β-catenin, and IL1ra. In the mice model, the co-injection of Lrp5- and/or IL1ra-overexpressing osteocytes induced a significant reduction in the tumor growth in the brain and maintained superior activity scores and brain damage scores. Furthermore, the minimally invasive administration of β-catenin-overexpressing CM to the head markedly reduced the size of brain tumors. In vitro analysis supported the in vivo observations, in which osteocyte-derived CM strikingly suppressed the tumorigenic behaviors of mammary tumor cells, and the overexpression of the three selected genes in osteocytes reduced the levels of tumorigenic genes such as IL1β, MMP9, Runx2, Snail, and TGFβ in breast cancer cells. Importantly, β-catenin-overexpressing CM upregulated TRAIL and cleaved caspase 3 for stimulating apoptosis, reduced chemokines such as CXCL1 and CXCL5 for blocking chemotaxis, and elevated tumor suppressors such as LIMA1, DSP, and p53. Most notably, extracellular histone H4 was elevated in osteocyte-derived CM by overexpressing Lrp5, β-catenin, and IL1ra, and the recombinant histone H4 acted as an atypical tumor suppressor. Collectively, the present study provided several lines of evidence to support the osteocyte-based therapeutic option to suppress brain tumors in a minimally invasive way. 

In generating enhanced tumor-suppressing CM, the overexpression of Lrp5 and β-catenin in Wnt signaling, as well as IL1ra in inflammatory signaling was examined. Paradoxically, Lrp5 and β-catenin are considered tumor-promoting genes, and much of the research efforts are directed to inhibit Wnt signaling [16,17]. These genes can, therefore, be considered as a double-edged sword with the opposing dichotomous role in tumor cells and osteocytes. To the best of our knowledge, it is for the first time to show the possibility of tumor suppression by activating canonical Wnt signaling. Interestingly, the inhibitory role of osteocyte-derived CMs is more selective to tumor cells than non-tumor cells (Appendix A). IL1ra is known for its suppressive action against IL1-driven inflammation and it was thus selected to reduce pro-inflammatory responses that may turn on tumorigenic genes. Notably, the overexpression of Lrp5 in brain-residing astrocytes did not exert the tumor-suppressing capability, and, to the contrary, it transformed them into tumor-promoting agents. Astrocytes are the most common glial cells and they are reported to support the proliferation and invasion of breast cancer cells in the brain [42]. Collectively, the result showed that the enhancement of the anti-tumor capability by the overexpression of Lrp5 depended on the type of cells.

While the action of histone H4 was hypothesized as a potential inducer of apoptosis, its extensive regulatory effects on CXCL1 and CXCL5, as well as the other tumor suppressors such as LIMA1, DSP, p53, and Trail in osteocyte-derived CM, were unexpected (Figure 7J). Histone H4 is one of the most highly conserved proteins, critically important for the structural integrity and accessibility of chromatin, and it is mostly located inside the nucleus to form octameric nucleosomes [32]. Interestingly, extracellular histone H4, which is a byproduct of cell death, is reported to serve as a warning signal and contribute to activating innate immune responses [43]. While the mechanism of this activation is not fully understood, the positive charge of histones may promote interactions with immune cells and downstream signaling. We observed the anti-tumor action of histone H4 not only in breast cancer cells but also in prostate cancer cells (data not shown). Further studies should clarify any linkage of histone H4-mediated immune responses to tumor-suppressing actions. Also, it is recommended to examine whether other histones may act as tumor suppressors in the extracellular domain and their action is restricted to a particular type of cancer. 

One of the major obstacles to the treatment of a brain tumor is the blood-brain barrier [44]. The transport analysis together with ex vivo, and in vivo data collectively provided multiple lines of evidence that the minimally invasive injection of CM is effective to deliver tumor-suppressing factors in the mouse model. The administered CM was processed using a filter with a 3 kDa cut off, and thus effective tumor-suppressing factors were 3 kDa or heavier. Of note, the molecular mass of histone H4 is 11.2 kDa. The human skull consists of the external compact bone and the internal trabecular bone [45], and this porous structure is similar to that of the mouse skull [46]. The tumor-suppressing result in response to CM indicated that CM can penetrate the brain. Further analysis is necessary to quantitatively evaluate the transport process.

There are several limitations in this study. First, we employed the mice models of tumor direct injection, which are different from the fashion of systemic tumor metastasizing. Second, the administration dose and concentration should be different in rodents and humans. MSCs were shown to present the anti-tumor capability by the overexpression of Lrp5 but astrocytes did not show any beneficial actions. Further analysis may evaluate if any other types of cells are the most appropriate secretary cells for inducing the tumor-suppressing capability.

## 4. Materials and Methods

### 4.1. Cell Culture

EO771 mammary tumor cells (CH3 BioSystems, Amherst, NY, USA) [47] and astrocytes (Cell Biologics, Chicago, IL, USA) were grown in DMEM (Corning, Inc., Corning, NY, USA), MLO-A5 osteocyte-like cells (obtained from Dr. L. Bonewald at Indiana University, Indianapolis, IN, USA), and bone marrow-derived MSCs (harvested from C57BL/6 mice) were cultured in αMEM (Gibco, Carlsbad, CA, USA). The culture media were supplemented with 10% fetal bovine serum and antibiotics (50 units/mL penicillin, and 50 µg/mL streptomycin; Life Technologies, Carlsbad, CA, USA), and cells were incubated at 37 °C with 5% CO_2_. To test the effect of TGFβ, EO771 cells were treated with TGFβ (500 ng/mL, Thermo Fisher Scientific, Waltham, MA, USA) for 1 day.

CM was prepared from MLO-A5 osteocytes and astrocytes at ~80% confluence after 24 h incubation and an Amicon filter unit with a cutoff mass at 3 kDa (Sigma-Aldrich, St. Louis, MO, USA) was used to remove microparticles and condense it by 10-fold. Cellular proliferation was examined using an MTT assay, and a wound-healing scratch assay and a Transwell invasion assay were conducted to evaluate cell motility and invasion capability, respectively, using the procedure previously described [48]. 

### 4.2. Spheroid Assay, Plasmid Transfection, and Western Blot Analysis

For the spheroid assay, 1.0 × 10^4^ cells/well were cultured in a U-bottom low-adhesion 96-well plate (S-Bio, Hudson, NH, USA) for 24 h. We transfected Lrp5 plasmids (#115907, Addgene, Watertown, MA, USA), β-catenin plasmids (#31785, Addgene), and IL-1ra plasmids (RG218518, Origene, Rockville, MD, USA) to MLO-A5 osteocytes and Snail plasmids (#31697, Addgene) to EO771 mammary tumor cells. All plasmids were transfected to approximately 2 × 10^6^ cells using lipofectamine 3000 (Life Technologies), and pcDNA was used as a control plasmid. In western blot analysis, cells were lysed in a radio-immunoprecipitation assay buffer and isolated proteins were size-fractionated and electro-transferred. We used antibodies against Snail, TGFβ, Lrp5, Runx2, Caspase 3 (cas3), cleaved Caspase 3 (c-cas3), histone H4 (Cell Signaling, Danvers, MA, USA), p53, CXCL2, IL1β, IL1ra (Invitrogen, Carlsbad, CA, USA), MMP9 (Santa Cruz Biotechnology, Dallas, TX, USA), TRAIL (Novus Biologicals, Centennial, CO, USA), LIMA1 (Novus Biologicals), DSP (Proteintech, Rosemont, IL, USA), CXCL5 (Abcam, Cambridge, MA, USA), and β-actin (Sigma-Aldrich). The uncropped gel images are shown in the Appendix A. RNA interference was conducted to silence histone H4 (Hist4h4 Select, Life Technologies), together with a nonspecific negative control siRNA (Silencer Select #1, Life Technologies). Cells were transiently transfected with siRNA with lipofectamine (Life Technologies), and the medium was replaced by a regular culture medium after 24 h.

### 4.3. Skull Diffusion Assay

The usage of human breast cancer tissues was approved by the Indiana University Institutional Review Board, and informed consent for research use was obtained from all patients. The experiment was performed following Indiana University’s human research protection program policies. A breast cancer tissue (~ 1 g; ER/PR+, HER2+), received from Simon Cancer Center Tissue Procurement Core, was manually fragmented with a scalpel into small pieces. Mouse skulls were isolated and rinsed with PBS. They were placed in a 24-well plate with the head top facing the well surface and tumor cells or breast cancer tissue fragments were grown. The well was filled with the control medium or osteocyte-derived CM. Fluphenazine (Sigma-Aldrich), a dopamine receptor antagonist that suppresses the growth of mammary tumor cells [20], was employed as a positive control to suppress tumor growth.

### 4.4. Animal Model

The procedures were approved by the Indiana University Animal Care and Use Committee and complied with the Guiding Principles in the Care and Use of Animals endorsed by the American Physiological Society. The sample size was decided using power analysis and stratified randomization was applied based on body weight. The mice were sacrificed on day 14, whereas the mice that died before day 7 were excluded.

To evaluate the effects of osteocytes or osteocyte-derived CM on mammary tumors or tumor-invaded tibiae, C57BL/6 female mice (~8 weeks; 18 mice per group) were randomly divided into three cell-testing groups (placebo, and osteocyte injection with and without Lrp5 overexpression), as well as two CM-testing groups (placebo, and β-catenin-overexpressing osteocyte-derived CM). Ten mice received the injection of EO771 cells (3.0 × 10^5^ cells in 50 µL PBS) to the mammary fat pad on day 0, while eight mice to the proximal tibia. The cell-testing groups received the co-injection of osteocytes (1.5 × 10^5^ cells) on day 0, while the CM-testing groups were given daily 50 µL of 10-fold condensed β-catenin CM subcutaneously to the mammary fat pad or systemically from the tail vein from day 2 to 14. The placebo group received the same volume of PBS. 

To evaluate tumor progression in the brain, C57BL/6 female mice (~8 weeks; 8 mice per group) were divided into five cell-testing groups, including the placebo, osteocyte group, and Lrp5-, IL1ra, and Lrp5-/IL1ra-overexpressing osteocyte groups, as well as three CM-testing groups (placebo, osteocyte-derived CM with and without β-catenin overexpression). All mice received the stereotaxic injection of EO771 cells (1.0 × 10^4^ cells in 15 µL PBS) into the right side of the frontal lobe on day 0, using the method previously reported [24]. The cell-testing groups received the injection of osteocytes (3.0 × 10^4^ cells in 15 µL PBS) on days 0 and 7. The CM-testing groups received 50 µL of osteocyte-derived CM with and without β-catenin overexpression subcutaneously to the right parietal daily from day 2 to 14. The placebo group received the same volume of PBS. 

### 4.5. Activity Score and Brain Damage Score

The activity score was determined blindly using the method previously described with minor modifications [49]. The score was in the range of 0 to 10 (10 for most active), with the score contributions for general appearance (0 to 2), natural behavior (0 to 3), provoked behavior (0 to 3), and body conditions (0 to 2). The brain damage score in the range of 0 to 10 was determined blindly based on 5 phenotypic features (surface roughness, bleeding, cortex swelling, cortex asymmetry, and encephalomalacia). Each feature was scored from 0 to 2, with the best brain damage score of 0.

### 4.6. X-ray Imaging and Histology

A whole-body X-ray image was taken using the Faxitron radiographic system (Faxitron X-ray Co., Tucson, AZ, USA) [50]. The tibiae were scored blindly at levels 0 to 3 as a bone damage score, in which 0 = normal; 1 = clear bone boundary with slight periosteum proliferation; 2 = bone damage and moderate periosteum proliferation; and 3 = severe bone erosion [51]. Brain samples were fixed in 4% paraformaldehyde in PBS for 24 h. They were dehydrated through a series of graded alcohols, cleared in xylene, and embedded in paraffin. The samples were sliced coronally with 4.5 µm thickness and H&E staining was conducted and analyzed in a blinded fashion.

### 4.7. Whole-Genome Proteomics

Proteins in CM were analyzed in the Dionex UltiMate 3000 RSLC system combined with a Q-exactive high-field hybrid quadrupole orbitrap mass spectrometer (Thermo Fisher Scientific). Proteins were first digested on-beads using trypsin/LysC. Digested peptides were desalted and separated using a trap and 50-cm analytical columns. Raw data were processed using MaxQuant (v1.6.3.3) against the Uniprot mouse protein database at a 1% false discovery rate allowing up to 2 missed cleavages. MS/MS counts were used for relative protein quantitation and proteins identified with at least 1 unique peptide and 2 MS/MS counts were considered for the final analysis. To evaluate the tumor-suppressing capability of the predicted candidates, we employed seven recombinant proteins such as heat shock protein family A member 8 (Hspa8), vimentin (Vim), (NBP1-30278, NBP2-35139; Novus, Littleton, CO, USA), heat shock protein 90 alpha family class B member 1 (Hsp90ab1) (OPCA05157; Aviva System Biology, San Diego, CA, USA), histone H4, ubiquitin (Ubc), peptidylprolyl isomerase A (Ppia), filamin A (Flna), and nucleolin (Ncl) (MBS2097677, MBS2029484, MBS286137, MBS962910, MBS146265; MyBioSource, San Diego, CA, USA). In the MTT assay, 5 μg/mL of each of these recombinant proteins were added and the viability of tumor cells was evaluated. The responses to histone H4 recombinant proteins were further evaluated using assays for EdU-based proliferation, and scratch-based motility, as well as western blot analysis for evaluating the levels of IL1β, Runx2, MMP9, TGFβ, and Snail in EO771 mammary tumor cells. 

### 4.8. Statistical Analysis

The data were expressed as mean ± S.D. Two independent samples and paired-samples were analyzed using Student’s t-test and paired t-test, respectively. For three independent experiments, statistical significance was evaluated using a one-way analysis of variance (ANOVA). Post hoc statistical comparisons among the groups were performed using Bonferroni correction with statistical significance at *p* < 0.05. The single and double asterisks in the figures indicate *p* < 0.05 and *p* < 0.01, respectively.

## 5. Conclusions

This work demonstrates the potential benefit of osteocytes and their CM for the treatment of brain tumors. While our experiments used a mouse mammary tumor cell line (triple-negative) as well as a human breast cancer tissue (positive in ER/PR and HER2), the response to osteocytes and their CM may depend on cancer cell types [27,52]. In summary, this study revealed that the local administration of β-catenin-overexpressing CM is capable of inhibiting the growth of brain tumors in a minimally invasive manner and extracellular histone H4 acts as an atypical tumor suppressor. The result suggests a novel CM-based therapeutic option to restrain breast cancer-associated tumor growth in the brain.

## Figures and Tables

**Figure 1 cancers-13-01061-f001:**
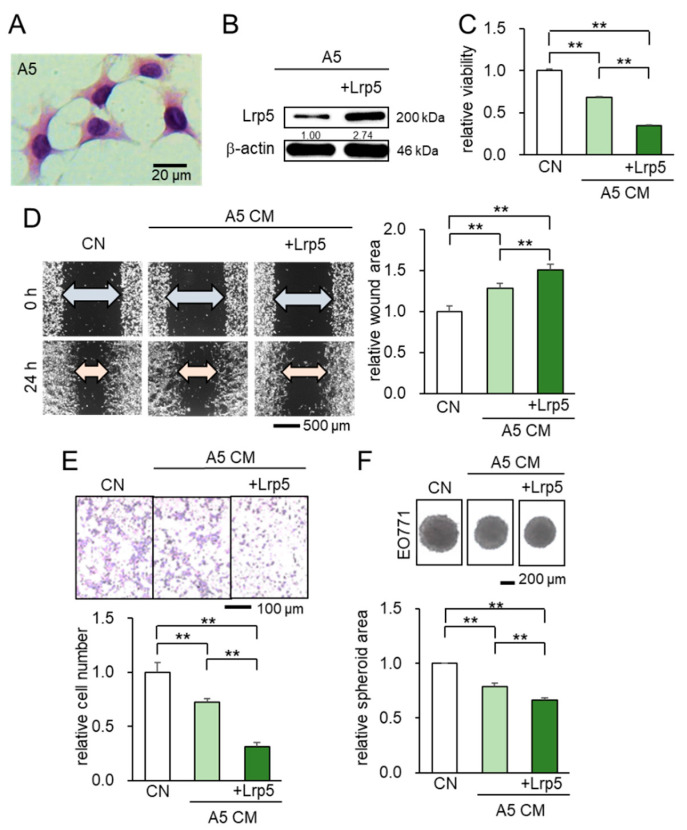
Suppression of the cellular proliferation, invasion, and migration in EO771 tumor cells by osteocyte-derived CM with and without Lrp5 overexpression. Of note, CN = control and CM = conditioned medium. The double asterisk indicates *p* < 0.01. The error bar indicates a standard deviation. (**A**) Representative images of MLO-A5 osteocytes (hematoxylin stained). (**B**) Transfection of Lrp5 plasmids in A5 osteocytes. (**C**–**E**) Reduction in MTT-based cellular viability, scratch-based migration, and Transwell invasion by osteocyte-derived CM. (**F**) Shrinkage of tumor spheroids in response to osteocyte-derived CM.

**Figure 2 cancers-13-01061-f002:**
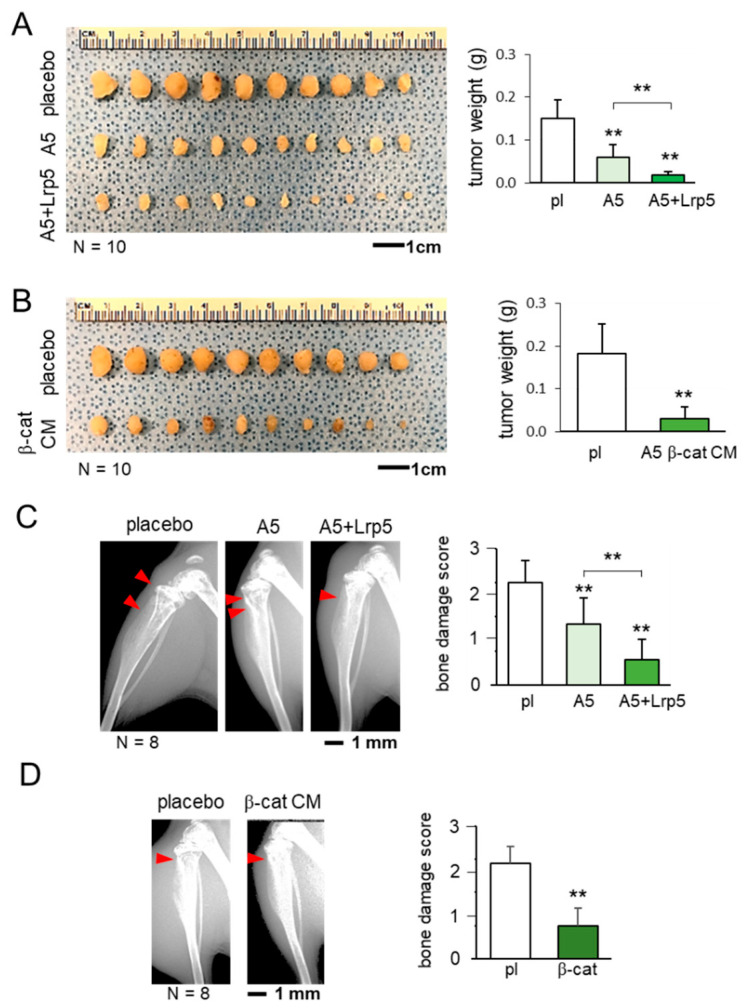
Suppression of the growth of mammary tumors and tumor-driven osteolysis by osteocytes and their conditioned medium. The double asterisk indicates *p* < 0.01. The error bar indicates a standard deviation. pl = placebo, and CM = conditioned medium. (**A**) Inhibition of the growth of mammary tumors by the co-injection of osteocytes and Lrp5-overexpressing osteocytes. (**B**) Reduction in the size of mammary tumors by the administration of b-catenin-overexpressing osteocyte-derived CM. (**C**) Prevention of bone loss in the proximal tibia by the co-injection of osteocytes and Lrp5-overexpressing osteocytes. The red arrowheads indicate the destructed area by tumor cells. (**D**) Protection of bone by the administration of b-catenin-overexpressing osteocyte-derived CM. The red arrowhead indicates the destructed area by tumor cells.

**Figure 3 cancers-13-01061-f003:**
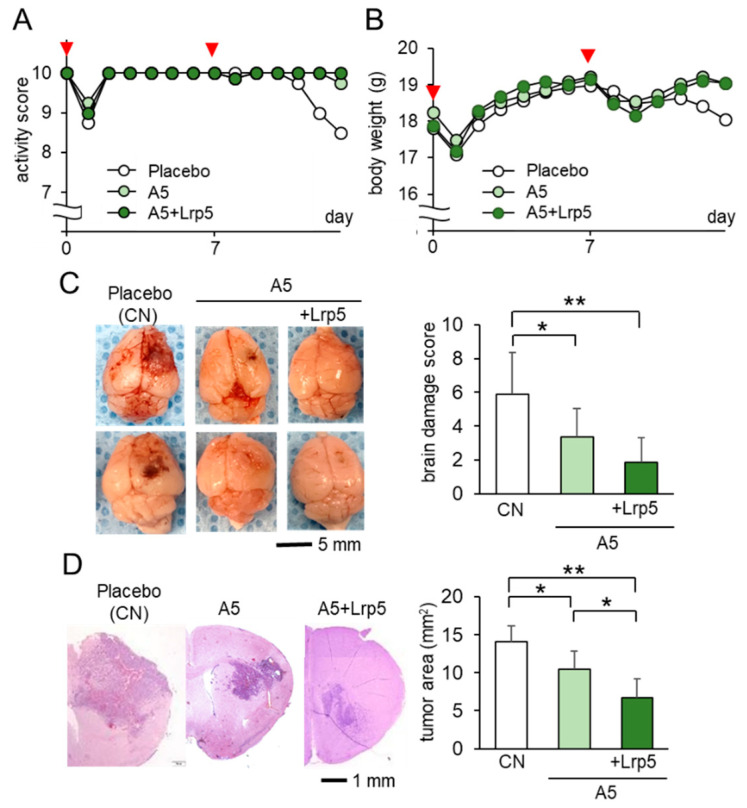
Suppression of the growth of brain tumors by the co-injection of Lrp5-overexpressed osteocytes in C57BL/6 mice (*N* = 8 per group). Of note, CN = control. The single and double asterisks indicate *p* < 0.05 and *p* < 0.01, respectively. The error bar indicates a standard deviation. (**A**,**B**) Activity score and body weight for 2 weeks. The red arrowheads indicate osteocyte injections. (**C**) Representative images of the whole brain with the brain damage score. (**D**) Histological images and the comparison of tumor areas in the coronal section.

**Figure 4 cancers-13-01061-f004:**
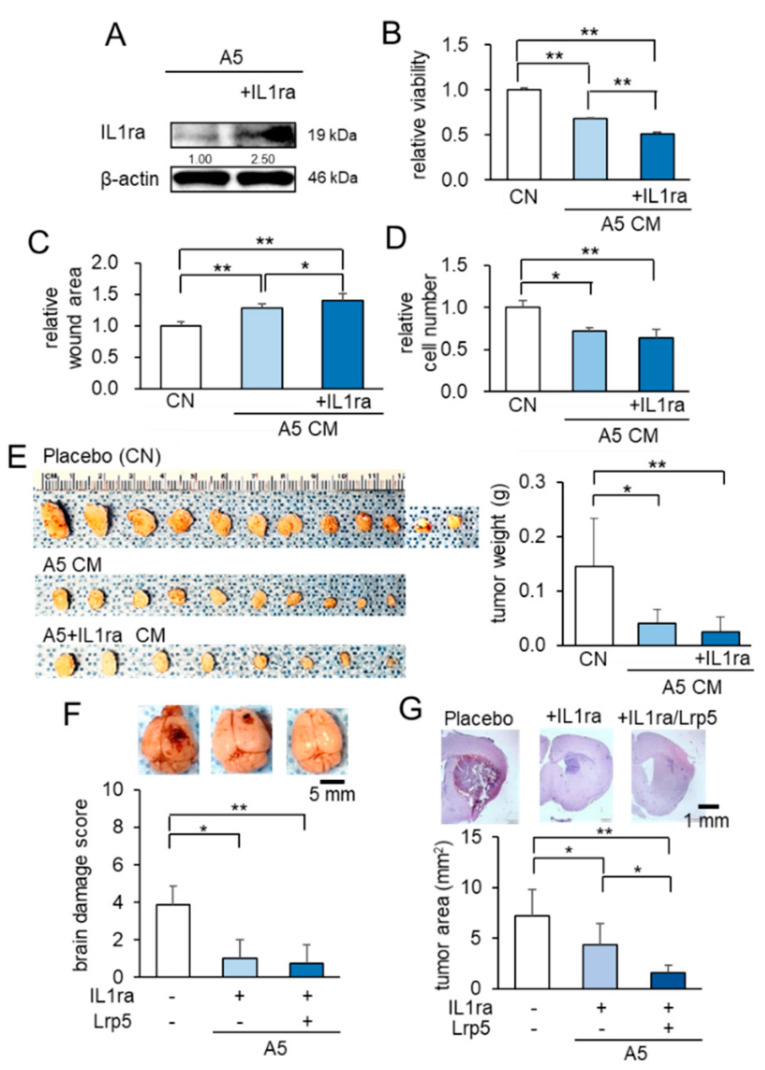
Anti-tumor effects of IL1ra overexpression in osteocytes. Of note, CN = control, and CM = conditioned medium. The single and double asterisks indicate *p* < 0.05 and *p* < 0.01, respectively. The error bar indicates a standard deviation. (**A**) Overexpression of IL1ra in osteocytes by the transfection of IL1ra plasmids. (**B**–**D**) Reduction in MTT-based cellular viability, scratch-based migration, and Transwell invasion in EO771 cells by IL1ra-overexpressing CM. (**E**) Reduction in the size and weight of mammary tumors in C57BL/6 mice (*N* = 8–12 per group) by the co-injection of IL1ra-overexpressing osteocytes. (**F**) Suppressive effects of osteocytes, overexpressed with IL-1ra, and/or Lrp5 in C57BL/6 mice. Whole brains in the three groups (placebo, injection of IL1ra-overexpressing osteocytes, and injection of IL1ra/Lrp5-overexpressing osteocytes; *N* = 8 per group). (**G**) Histological analysis, showing that the osteocyte injection significantly suppressed tumor growth, and the overexpression of IL1ra and Lrp5 further reduced tumor progression.

**Figure 5 cancers-13-01061-f005:**
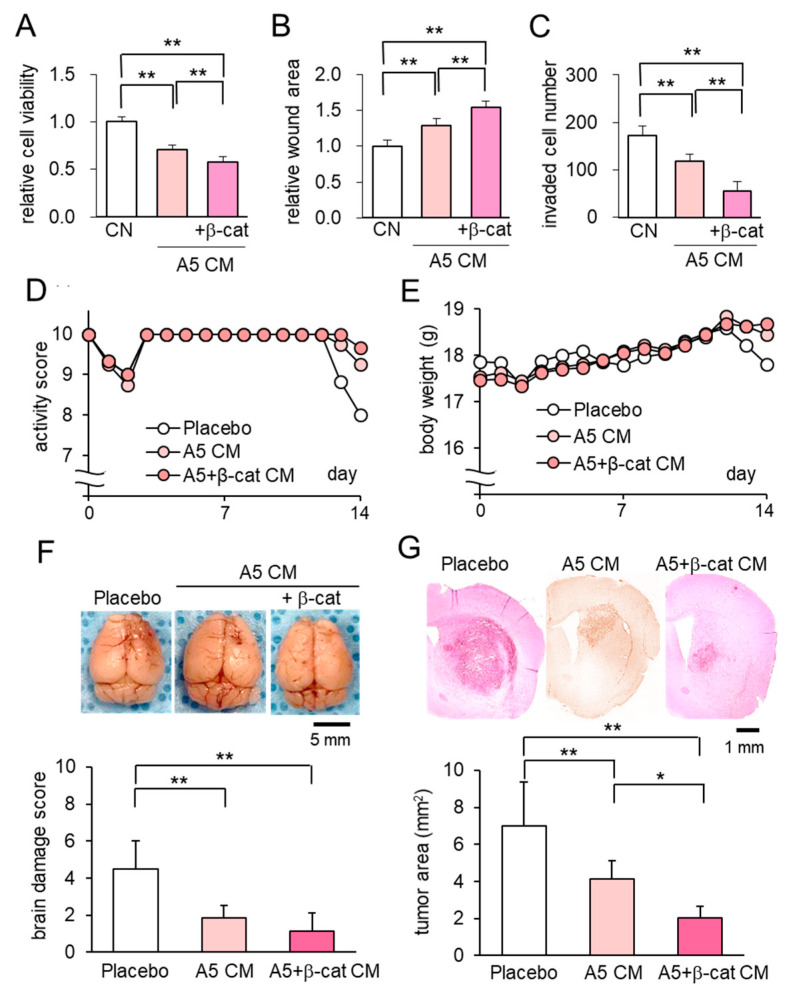
Anti-tumor effect of osteocyte-derived CM with β-catenin overexpression on brain tumors. Of note, β-cat = β-catenin, CN = control, and CM = conditioned medium. The double asterisks indicate *p* < 0.01. The error bar indicates a standard deviation. (**A**–**C**) Reduction in MTT-based cell viability, scratch-based migration, and Transwell invasion in EO771 cells by β-catenin overexpressing CM. (**D**,**E**) Activity score and body weight for 2 weeks (*N* = 8 per group). (**F**) Whole brains in the three groups (placebo, and the injections of A5 CM and β-catenin overexpressing A5 CM) with the brain damage score. (**G**) The single and double asterisks indicate *p* < 0.05 and *p* < 0.01, respectively. Histological analysis, showing that osteocyte-derived CM with the overexpression of β-catenin significantly suppressed tumor growth.

**Figure 6 cancers-13-01061-f006:**
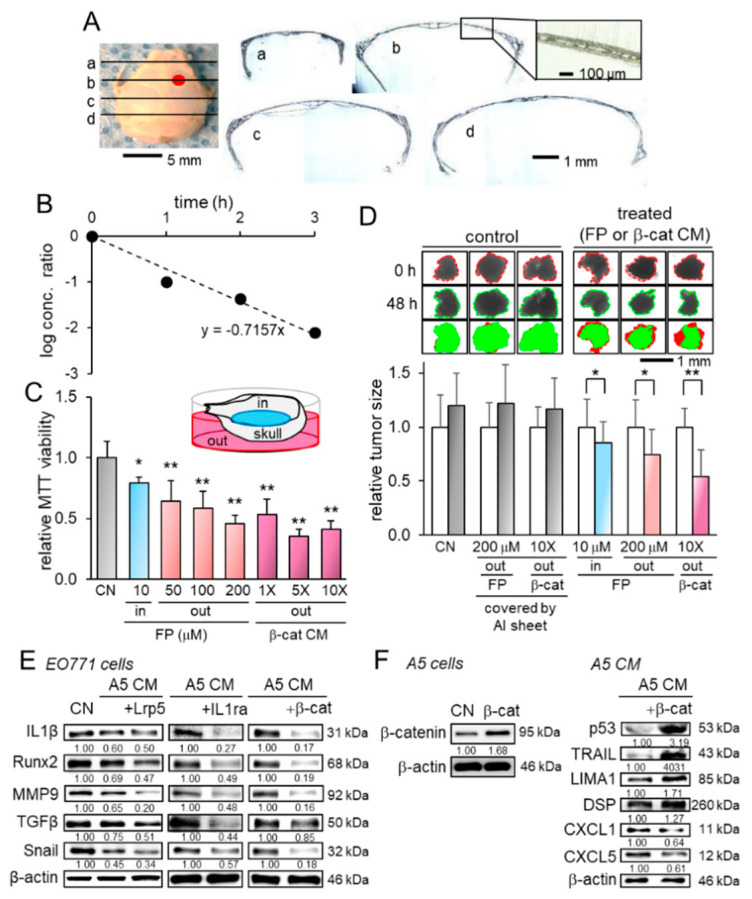
Transport analysis of conditioned medium through the skull and the alterations in gene expression. Of note, FP = fluphenazine, β-cat = β-catenin, CN = control, and CM = conditioned medium. The single and double asterisks indicate *p* < 0.05 and *p* < 0.01, respectively. The error bar indicates a standard deviation. (**A**) Coronal sections of the skull. The red mark on the skull image indicates the injection site of EO771 tumor cells. (**B**) Temporal changes in the diffusion-linked ratio (see the derivation in the Appendix A). (**C**) Transport analysis with MTT-based viability, in which EO771 cells were cultured on the inside of the skull. FP and β-catenin-overexpressing CM in the outside of the skull decreased the viability of tumor cells on the inside of the skull. Of note, 10 µM of FP (an antagonist of dopamine D2 receptor) in the inside of the skull was employed as a positive control for tumor suppression. (**D**) Changes in the size of human mammary tumor fragments. In the control groups, the transport of FP and CM was blocked by the coverage of the skull with Aluminum foil. The administration of FP and CM in the outside of the skull shrunk the mammary tumor fragments in the inside of the skull. Of note, 10 µM of FP in the inside of the skull was employed as a positive control for tumor suppression. The left and right bars indicate the relative tumor size after 0 and 48 h, respectively. (**E**) Reduction in IL1β, Runx2, MMP9, TGFβ, and Snail in EO771 cells by osteocyte-derived CM with and without the overexpression of Lrp5, IL1ra, and β-catenin. (**F**) Overexpression of β-catenin in osteocytes, and the elevation in p53, TRAIL, LIMA1, and DSP, as well as the reduction in CXCL1 and CXCL5 in osteocyte-derived CM by the overexpression of β-catenin.

**Figure 7 cancers-13-01061-f007:**
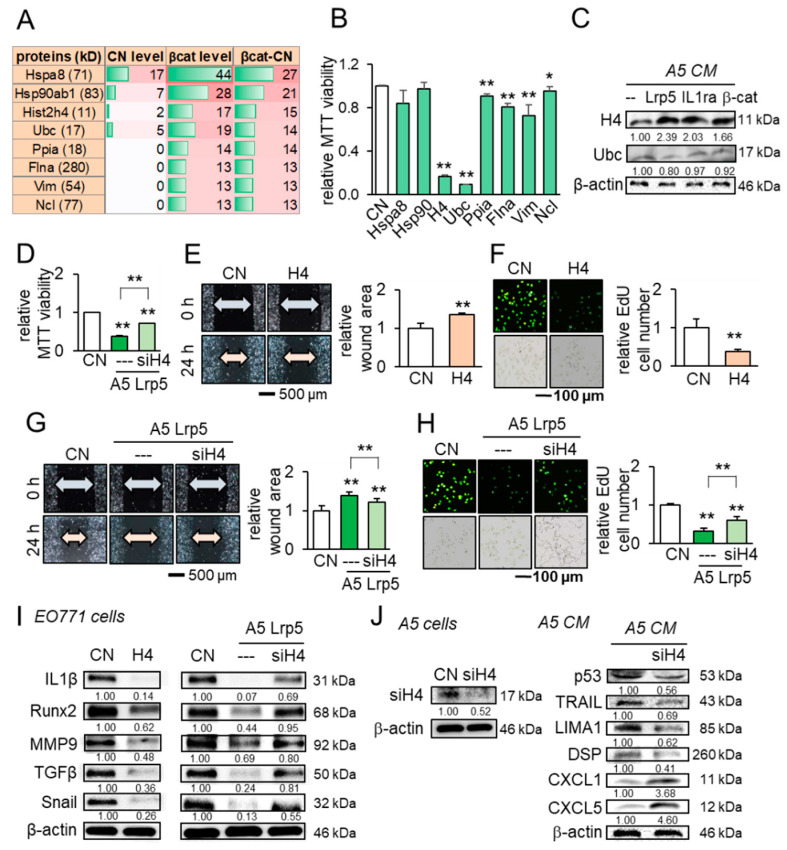
Anti-tumor actions of osteocyte-derived conditioned medium and the tumor-suppressing role of histone H4 in EO771 mammary tumor cells. CN = control, CM = conditioned medium, H4 = histone H4, and siH4 = histone H4 siRNA. The single and double asterisks indicate *p* < 0.05 and *p* < 0.01, respectively. The error bar indicates a standard deviation. (**A**) Eight tumor-suppressing protein candidates, predicted by whole-genome proteomics. (**B**) Effects of the predicted candidates on the MTT-based viability of EO771 mammary tumor cells. (**C**) Elevation of histone H4 in osteocyte-derived CM by the overexpression of Lrp5, IL1ra, and β-catenin. (**D**) Reduction in MTT-based viability by histone H4 and its suppression by histone H4 siRNA. (**E**,**F**) Reduction in the scratch-based migration and EdU-based proliferation by histone H4. (**G**,**H**) Suppression of the reduced migration and proliferation by histone H4 siRNA. (**I**) Reduction in IL1β, Runx2, MMP9, TGFβ, and Snail by histone H4, and the suspension of their reduction by histone H4 siRNA. (**J**) Silencing of histone H4, and Reduction in p53, TRAIL, LIMA1, and DSP, as well as the elevation of CXCL1 and CXCL5 in osteocyte-derived CM by histone H4 siRNA.

**Figure 8 cancers-13-01061-f008:**
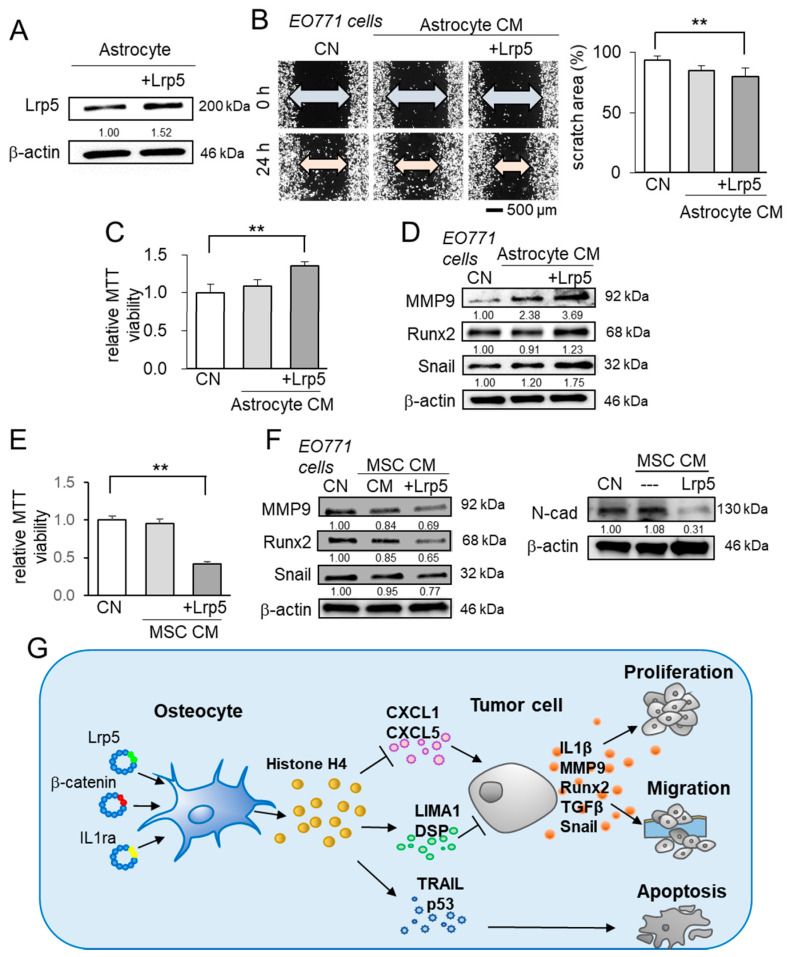
Characterization of CMs derived from astrocytes and mesenchymal stem cells, and the regulatory mechanism of osteocyte-driven tumor suppression. The double asterisk indicates *p* < 0.01. The error bar indicates a standard deviation. CN = control, and CM = conditioned medium. (**A**–**C**) Scratch-based migration and MTT-based viability of EO771 cells in response to astrocyte-derived CM with and without Lrp5 overexpression. (**D**) Elevation in MMP9, Runx2, and Snail in EO771 cells by astrocyte-derived CM. (**E**) MTT-based viability of EO771 cells in response to MSC-derived CM with and without Lrp5 overexpression. (**F**) Reduction in MMP9, Runx2, and Snail in EO771 cells by MSC-derived CM, and the reduction of N-cad (N-cadherin) in EO771 cells by Lrp5-overexpressing MSC CM. (**G**) Schematic illustration of the regulatory mechanism of the anti-tumor action of osteocytes. Osteocytes have the potential to suppress the expression of IL1β, MMP9, Runx2, TGFβ, and Snail in tumor cells. The overexpression of Lrp5, β-catenin, and IL1ra strengthens their anti-tumor capability. In particular, the overexpression of β-catenin elevated extracellular histone H4 that served as a tumor suppressor. The elevated histone H4 downregulated the tumor promotors (CXCL1 and CXCL5) and upregulated the tumor suppressors and apoptosis inducers (LIMA1, DSP, TRAIL, c-cas3, and p53).

## Data Availability

The data presented in this study are available in this article (and Appendix A).

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
