# Peer review of "Inhibition of the Growth of Breast Cancer-Associated Brain Tumors by the Osteocyte-Derived Conditioned Medium"

_cancers, 2021, doi:10.3390/cancers13051061_

Round 1

Reviewer 1 Report

The authors revealed that brain tumor cell's metastasis is suppressed by osteocytes, especially conditioned media from osteocytes. Lrp5, IL1ra, and beta-catein in osteocytes are critical for anti-tumor effects. Moreover, these genes suppressed MMP9, Runx2, TGFbeta and Snail and enhanced TRAIL and p53. Histone H4 is also important as an anti-tumor effetcs. They are aimed to establish a new way to inhibit brain metastasis by treatment of conditioned media from manipulated osteocytes.

In several parts, including abstract, it's not clear in which the molecular expression is upregulated or suppressed. For instance, "the overexppression of the above genes downregulated oncogenic genes such as MMP9, Runx2, TGFβ, and Snail." in abstract. Which cells express MMP9,...,Snail? Please write them clearly.

Does the conditioned media from osteocytes have an effect to normal cells? I wonder whether the CM shows bad side-effects when we use the CM for actual therapy. You should show the effect of the CM on normal cells, especially nerve cells and blood vessel cells.

I couldn't see the "red arrowhead" in figure 2C, D.

Please show which reagent you used for plasmid transfection in Materials and Methods section.

I couldn't find uncropped gel images in supplementary information.

I think in Figure 5F, there are 3 groups but the legend said 2 groups.

In Figure6D, what is the difference of right and left bars? Please write in the legend.

In Figure6, you showed the CM can go through the skull. How about the brain itself? Probably cancer cells on the brain surface will be affected by the CM, but cancer cells inside of the brain may not be affected the CM through the skull.

In the section 2.7, you said "In the MTT assay, the inhibitory action of histone 207H4 is stronger to tumor cells than to non-tumor cells (Suppl. Fig. 4" but I couldn't find it.

How to prepare extracellular histone H4 and treat the cells with extracellular histone H4? Please describe it.

Do you show CXCL1, CXCL5, LIMA1, and DSP actually regulate IL1b, MMP9, Runx2, TGFb, Snail in cancer cells? Also, do IL1b, MMP9, Runx2, TGFb, Snail affect proliferation and migration in cancer cells? At least about several molecules, you should perform experiments.

In discussion, you said "Lrp5 was selected to be overexpressed since we previously observed with breast cancer cells that osteocytes with a higher expression level of Lrp5 were more potent to suppress tumor progression". Please mention about it in introduction too. Otherwise, we don't know why you focused on this molecule. it should be written about IL1ra and b-catenin, too.

Reviewer 2 Report

Sano et al in their manuscript titled “Inhibition of the growth of breast cancer-associated brain tumors by the osteocyte-derived conditioned medium“ examined a therapeutic option with a conditioned medium (CM), focusing on the role of Lrp5 and β-catenin in Wnt signaling, and IL1ra in osteocytes. The author found that osteocytes presented the innate anti-tumor effect and the overexpression of the above genes strengthened their action. The authors injected the conditioned medium into mouse models and showed inhibition of mammary tumors and tumor-driven osteolysis, importantly, Lrp5- and/or IL1ra-overexpressing osteocytes or the local administration of β-catenin-overexpressing CM markedly inhibited brain tumors. They showed that the overexpression of the above genes downregulated oncogenic genes such as MMP9, Runx2, TGFβ, and Snail. β-catenin overexpression also downregulated CXCL1 and CXCL5 and upregulated tumor suppressors such as LIMA1, DSP, p53, and TRAIL. By using proteomics they revealed that histone H4 was enriched in CM and acted as an atypical tumor suppressor. Lrp5-overexpressing MSCs were also shown to act as anti-tumor agents. Over all the authors demonstrated the therapeutic role of engineered CM in brain tumors and the tumor-suppressing action of extracellular histone H4 shedding light on the potential of CM-based therapy for breast cancer-associated brain metastases in a minimally invasive manner.

The topic is interesting and well planned. However I have few points to be addressed.

The authors should italicize In vivo and In vitro in the manuscript wherever they used these terms.

The authors claim that B-catenin is tumor suppressive. In general B-catenin is known to promote tumorigenesis in many cancers by activating Wnt signaling down stream targets like c-Myc, cyclin D1 etc. Did the authors measure how these genes regulate in their studies or how can they justiy this counter finding that OE of CTNNB1 leads to reduced tumor growth?

For figure 6 the authors should add B-catenin overexpression in cells which were treated in Western blot.

Similarly for figure 7, the authors should show H4 down regulation for siH4 in western blot and Lrp5 for figure 8

Reviewer 3 Report

The manuscript demonstrates the therapeutic role of engineered CM in brain tumors and the tumor-suppressing action of extracellular histone H4. The result sheds light on the potential CM-based therapy for breast cancer-associated brain metastases in a minimally invasive manner. The studies are nicely executed, and most of the findings are straight forward. While the concept of the work is very interesting and informative, there are a few major concerns that the authors need to address in their study. 

  1. Please provide significance of the research in more details.
  2. Please improve method section
  3. Please improve discussion section
  4. Please provide more representative images for Fig 7F and H
  5. Please add immunoblot for EMT markers like fibronectin, ecadherin, zeb 1, Zo-1, N-cadherin, slug, Vimentin in figure 6, 7 and 8
  6. Provide densitometric analysis for all immune blots in the manuscript
  7. Please cite the following articles in your manuscript and include in reference section
    1. Transglutaminase-2 facilitates extracellular vesicle-mediated establishment of the metastatic niche
    2. Spleen Tyrosine Kinase–Mediated Autophagy Is Required for Epithelial–Mesenchymal Plasticity and Metastasis in Breast Cancer.
    3. Autocrine fibronectin inhibits breast cancer metastasis.
    4. Pyruvate carboxylase supports the pulmonary tropism of metastatic breast cancer
    5. Regulation of epithelial-mesenchymal transition and metastasis by TGF-β, P-bodies, and autophagy
    6. The Dynamic Relationship of Breast Cancer Cells and Fibroblasts in Fibronectin Accumulation at Primary and Metastatic Tumor Sites
    7. Fibronectin expressing mesenchymal tumor cells promote breast cancer metastasis
    8. Dynamic transition of the blood-brain barrier in the development of non-small cell lung cancer brain metastases
    9. Inhibition of pyruvate carboxylase by 1α, 25-dihydroxyvitamin D promotes oxidative stress in early breast cancer progression

Reviewer 4 Report

Sano et al present a comprehensive analysis on the anti-cancerous effects of osteocyte medium addition to breast cancer cells and in mice model. The rationale for the analysis of a tumor-osteocyte interaction in terms of cerebral metastasis remains elusive and the clinical relevance is questionable. Since breast cancer metastasis occurs frequently in the bone a osteocyte-bone metastasis model appears more suitable for studying this interaction. Overall the manuscript is well written but appears overloaded.

Line 24/25, sentance unclear

2.1 The effects of the injected CM are displayed by tumor osteolysis and metastatis but in my opinion a systemic effect must be ruled out ie. Displaying the mice weigh activity socere and body weight before and after treatment as in figure 3 same comment applies to 2.3 figure 4

The results and discussion on extracellular Histone 4 remains elusive. Since the presence of intranuclaer DNA is highly proapoptotic itselve, the H4 induced mechanisms proposed should be discussed in this context .

A detailed limitation section should be stated in the discussion

Round 2

Reviewer 1 Report

In Discussion, the authors mentioned that "Interestingly, the inhibitory  role  of  osteocyte-derived  CMs  is  more  selective  to  tumor  cells  than  non-tumor cells  (data  not  shown)". Please show your data as supplemental figures.

In Materials and Methods, the authors mentioned "All plasmids weretransfected to approximately 2x106cellsusing lipofectamin (Life Technology)". Please show which lipopfectamine (3000, 2000, or another one) you used.

In cover letter, the authors mentioned that "the tumor-suppressing result in response to CM indicated that CM can penetrate the brain. We agree that further analysis is necessary to quantitatively evaluate the diffusion process". Please describe it in Discussion section.

You mentioned that "In the revision, we added the new Suppl. Fig. 4 (responses to TGFb and Snail overexpression in EO771 cells)" and "Based on the revision time for 10 days, we focused on the responses to TGF recombinant protein and the overexpression of Snail in EO771 cells and conducted MTT-based viability and scratch-based motility assays (new Suppl. Fig. 4)". However, I could not find the data in supplementary figure. Please submit with the file again.

Reviewer 3 Report

The authors have addressed all the comments.

Author Response

We appreciate for your review and valuable comments.

Reviewer 4 Report

The authors have addressed my concerns adequately. My reservations are resolved.

Author Response

(The authors gave the same response as above.)
